# Effect of Omega-3 Polyunsaturated Fatty Acid Supplementation on Clinical Outcome of Atopic Dermatitis in Children

**DOI:** 10.3390/nu16172829

**Published:** 2024-08-24

**Authors:** Tena Niseteo, Iva Hojsak, Suzana Ožanić Bulić, Nives Pustišek

**Affiliations:** 1Referral Center for Pediatric Gastroenterology and Nutrition, Children’s Hospital Zagreb, 10000 Zagreb, Croatia; 2Faculty of Food Technology and Biotechnology, University of Zagreb, 10000 Zagreb, Croatia; 3Department of Pediatrics, School of Medicine, University of Zagreb, 10000 Zagreb, Croatia; 4Department of Pediatrics, School of Medicine, University J.J. Strossmayer, 31000 Osijek, Croatia; 5Department of Pediatric and Adolescent Dermatology and Venerology, Children’s Hospital Zagreb, 10000 Zagreb, Croatia

**Keywords:** atopic dermatitis, children, omega-3 fatty acids, eicosapentaenoic acid (EPA), docosahexaenoic acid (DHA), gamma-linolenic acid (GLA)

## Abstract

The use of omega-3 fatty acids (omega-3 FA) in the treatment of atopic dermatitis (AD) is an area of ongoing research. Some studies suggest that dietary supplementation with omega-3 FA can help manage symptoms of AD by reducing lesion severity, skin inflammation, dryness and itching, while others show no significant beneficial effect. The aim of this study was to evaluate the effect of omega-3 FA from fish oil in combination with gamma-linolenic acid (GLA) from blackcurrant seed oil in children with AD. This is a longitudinal, prospective, randomized, triple blind, placebo-controlled parallel clinical trial. The study was conducted during the 2-year period throughout autumn, winter, and spring, avoiding the summer when AD usually improves. Children were randomized to receive the active study product (Mega Kid^®^) containing a specific blend of omega-3 and omega-6 fatty acids or placebo. The primary outcomes were changes in severity of AD measured using SCORing Atopic Dermatitis (SCORAD), patient-oriented SCORAD (PO-SCORAD) and the difference in topical corticosteroid (TCS) use. The secondary outcomes were changes in itch intensity, sleep quality and Family Dermatology Life Quality Index (FDLQI). Data were analyzed for 52 children (26 in the intervention group and 26 in the placebo group). In children receiving the active product, intention-to-treat analysis showed that after 4 months of treatment, there was a significant decrease in the SCORAD index (from median 42 to 25, *p* < 0.001) and the use of topical corticosteroids (from median 30 to 10 mg/month, *p* < 0.001), but also significant improvements in itch, sleep quality, and overall quality of life. Omega-3 fatty acids in combination with GLA and vitamin D may decrease symptoms and were associated with an improvement clinical picture of AD in children. Therefore, we can conclude that supplementation with this specific combination could be considered a safe and effective intervention that may significantly reduce the severity of AD in pediatric patients.

## 1. Introduction

Atopic dermatitis (AD) is an inflammatory, pruritic, chronic, or chronically relapsing skin disease. The complex pathogenesis of AD is a result of genetic predisposition, skin barrier dysfunction, immunological factors, skin microbiome dysbiosis, environmental factors, and strong psychosomatic influence [1]. The increasing incidence of AD in recent decades may be explained by changes in lifestyle and diet, in particular, a shift in the balance between dietary polyunsaturated fatty acids (PUFAs) with increased dietary intake of omega-6 fatty acids (omega-6 FA) intake [2]. The ideal omega-6 FA to omega-3 FA ratio appears to be from 1:1 to 6:1; however, the Western diet produces ratios of more than 25:1. This change in balance could influence inflammatory responses [2,3,4,5,6,7]. Omega-3 FA and omega-6 FA compete for the same enzymes during conversion into active metabolites. Omega-6 FA is a precursor of arachidonic acid (AA), a lipid with the potential for inflammatory pathway activation through the release of pro-inflammatory eicosanoids which can exacerbate the inflammatory processes underlying AD [2,3,8]. Omega-3 FA, alpha-linolenic fatty acid (ALA), is converted to eicosapentaenoic acid (EPA) and then to docosahexaenoic acid (DHA). Eicosanoids derived from EPA and DHA have high anti-inflammatory potential [2,8]. 

The high intake of omega-6 FA is known to enhance pro-inflammatory response by inducing the production of pro-inflammatory eicosanoids. Omega-6 FA, linoleic acid (LA), is converted to AA, the precursor of inflammatory prostaglandins and leukotrienes, while ALA, an omega-3 FA, is converted to EPA and subsequently to DHA. These FAs are known to be the most biologically potent precursors of anti-inflammatory mediators. However, LA, before it converts to AA, goes through a stage of gamma-linolenic acid (GLA) and dihomo-gamma-linolenic acid (DGLA), both involved in the induction of anti-inflammatory eicosanoids [9].

In patients with AD, a reduced enzyme activity of delta-6-desaturase was observed, leading to higher levels of LA and lower levels of GLA, resulting in lower concentrations of anti-inflammatory GLA and its metabolites [10,11]. This concept was supported by a Swedish study that showed elevated LA concentrations and, on the other hand, significantly reduced concentrations of DGLA, AA, and DHA [12]. Moreover, LA and GLA have an essential role in maintaining the integrity of the skin barrier. GLA has been shown to regenerate the human skin barrier, increase ceramide synthesis, and prevent excessive water loss [13,14]. This explains the importance of the balance in consuming omega-3 FA and omega-6 FA and their role in the pathogenesis of AD through effects on inflammation, immune modulation, and skin barrier function. This nutritional disbalance could contribute to the higher production of pro-inflammatory cytokines and elevate the risk of development of atopic diseases [15]. 

Most of the published studies on the role of omega-3 FA in AD demonstrated their pivotal preventive role in the context of atopic diseases and allergies; however, their role in the treatment of atopic diseases, e.g., AD, is still unclear [16,17,18]. 

The treatment of AD includes proper education, avoiding triggers (clinically relevant allergens and irritants), regular use of emollients, and topical anti-inflammatory medication (reactive or proactive). After topical treatments are unsuccessful in patients with moderate-to-severe disease, the next approaches typically include either phototherapy or systemic therapies such as conventional immunosuppressants, biologics, or small molecules like JAK inhibitors. The use of PUFA in the treatment of AD is an area of ongoing research. While some studies suggest potential benefits, the evidence is not yet strong enough to establish standardized guidelines for PUFA supplementation in the treatment of AD [19]. Studies suggest that dietary supplementation with omega-3 FA can help manage symptoms of AD by reducing lesion severity, skin inflammation, dryness, and itching [2]. 

Therefore, the aim of this study was to evaluate the effect of supplementation with omega-3 FA from fish oil in combination with GLA from blackcurrant seed oil in children with AD.

## 2. Materials and Methods

This is a longitudinal, prospective, randomized, triple-blind, placebo-controlled parallel clinical trial approved by the Ethics Committee of Children’s Hospital Zagreb. Written informed consent was obtained from the parent or guardian of the child participating in the study.

The study was conducted during the 2-year period throughout autumn, winter, and spring 2021 to 2023 to avoid summer when AD usually improves. 

### 2.1. Participants

The participants were pediatric patients diagnosed with AD and referred to a pediatric dermatology outpatient clinic. Children from 1 up to 8 years of age with AD were included. The diagnosis was based on the Rajka–Hanfin criteria [20]. Children with moderate to severe AD, according to the SOCORAD (SCORing Atopic Dermatitis) index, were eligible for the study [21]. SCORAD is calculated using a three-criteria questionnaire based on the involved area, intensity of eczema, and subjective symptoms (itching and sleeplessness). The total score ranges from zero to 103, where SCORAD 0 to 24 describes mild, 25 to 49 moderate, and >50 severe forms of AD. All children were enrolled by a pediatric dermatologist. 

The exclusion criteria were children younger than 1 year and older than 8 years of age, children with SCORAD ≤ 25, children receiving any other supplements except vitamin D, those on or in need of phototherapy and systemic therapy, and those allergic to fish. 

Randomization of the study participants was performed by using the Random Allocation Software 2.0. to generate randomized blocks for assigning patients to two groups of equal size where every patient had a number assigned and received the labeled investigational product or placebo successively.

The investigational product was a citrus-flavored syrup. Both preparations, active and placebo, were supplied by 4U Pharma Gmbh, Herisau, Switzerland. The company 4U Pharma had no involvement in the design, implementation, analysis, and interpretation of the data. Products were packed in identical dark brown bottles different only in labels (A or B). All participants, clinicians, data collectors, outcome adjudicators, and data analysts did not have access to the details of the group assignment.

### 2.2. Study Product and Administration

The active study product (Mega Kid^®^, 4U Pharma Gmbh, Herisau, Switzerland) contained fish oil, EPA, DHA, GLA, vitamin D3, and, as inactive ingredients, medium chain triglycerides (MCT oil), lemon-lime flavoring, and anhydrous citric acid. The placebo study product consisted of an identical formulation of inactive ingredients (MCT oil, lemon-lime flavor, and anhydrous citric acid).

Both of the products, active and placebo, were of the same taste, color and smell. Products were stored at temperatures below 25 °C, and the shelf life was 12 months. Once opened, the patients were instructed to store the product in the refrigerator while using it. Both the research staff and the patients were unaware of the nature of the product. The intervention period lasted 4 months. During the entire intervention period, the subjects were not allowed to consume any other supplements except vitamin D. All participants were followed by a telephone call or an email every 4 weeks. 

Children were assigned to one of the treatment groups (experimental or control) following a randomization procedure performed with computer-generated numbers. The daily dose of the investigational product was 5 mL, with the active product containing 2 g of fish oil, 600 mg EPA, 400 mg DHA, 10 mg GLA, and 5 mcg of vitamin D3. Both groups received conventional AD treatments, such as topical emollients and low-potency TCS (usually alclometasone dipropionate ointment or 40% and 60% alclometasone dipropionate emollient).

### 2.3. Outcomes

Primary outcomes were a change in the severity of AD and the difference in TCS use. Secondary outcomes were changes in itch intensity, sleep quality, and FDLQI.

Demographic parameters, medical history, use of medication and supplements, AD severity assessed by SCORAD and PO-SCORAD (patient-oriented SCORAD) [22], itch (assessed by numerical rating scale from 1 to 10, with 1 indicating low intensity and 10 highest intensity itch), sleep disturbance (using numerical rating scale 1 to 10, with 1 indicating best sleep quality and 10 the worst sleep quality), AD-related quality of life using the Croatian version of the Family Dermatology Life Quality Index (FDLQI) [23] were recorded at baseline and at the end of treatment period (4 months). 

The PO-SCORAD index is a self-assessment tool allowing parents to evaluate the severity of AD comprehensively by using subjective and objective criteria derived from the SCORAD. The FDLQI is a dermatology-specific instrument that measures the disease effect on the health-related quality of life (QOL) of family members of patients with skin diseases. The 10-item questionnaire is designed to assess the different aspects of the quality of life of parents, caregivers or partners of patients with various skin diseases, in particular emotional and physical well-being, relationships, social life, leisure activities, burden of care, effect on job or study, housework, and financial burden of the disease. Participants answer questions choosing from a 4-point scale: 0, not at all or not applicable; 1, a little; 2, a lot; 3, very much. The scores of individual items (0–3) are added to give a total score that ranges from 0 to 30; higher scores indicate poorer QOL. To obtain a valid Croatian version of the FDLQI, we followed the guidelines for the cross-cultural adaptation of health-related QOL measures [24]. The use of TCS was determined by the parent’s report on TCS use, according to the weight of the used amount of TCS per month (expressed as an absolute number). 

During the intervention period, data on tolerability and adverse events were collected. 

### 2.4. Sample Size

The sample size was calculated based on previous data that showed that the itching score reported by patients at the end of 3 months in the intervention group was 2.6 ± 2.7, and in the placebo group, the score was 5.2 ± 2.7 [25]. The effect size based on these results was equal to 1.03. Based on these results, the calculated sample size was 42 participants (alpha 0.05, power 0.80). In order to account for possible drop-outs, four additional patients were added. 

### 2.5. Statistics

Descriptive statistics were used to describe the basic features, including age, gender, duration of the intervention, and symptoms. The normality of the data distribution was analyzed with the Smirnov–Kolmogorov test. For the primary analyses, we evaluated changes in the severity of AD (SCORAD, PO-SCORAD) and the difference in TCS use. Due to non-normal distribution, the difference was analyzed by the nonparametric Mann–Whitney test and described as median (range).

For secondary outcomes, the changes in itch intensity, sleep quality and FDLQI were analyzed again by the use of the nonparametric Mann–Whitney test and described as median (range).

Two-sided *p*-values of less than 0.05 were considered to indicate statistical significance for the primary outcomes.

Statistical software SPSS (version 23; SPSS, Chicago, IL, USA) was used for all statistical analyses. All analyses were performed on the intention-to-treat basis, in which all of the participants in a trial were analyzed according to the intervention to which they were assigned.

## 3. Results

In this study, 53 patients were enrolled, with 52 patients completing the end-of-study follow-up visit as per protocol (26 in the intervention group and 26 in the placebo group) (Figure 1). 

There were no significant differences with respect to gender, age, food allergy, vitamin D supplementation, and severity of AD between the intervention and placebo groups (Table 1). 

Intention-to-treat analysis showed that after 4 months of treatment, the SCORAD index decreased significantly in the intervention group (from median 42 to 25, *p* < 0.001) but there was no change in the placebo group (from median 39 to 39, *p* = 0.795) (Figure 2). The median use of TCS in the intervention group was the same as in the placebo group (30 mg/month). After 4 months of the intervention period, a significant decrease was detected in TCS use in the intervention group (30 to 10 mg/month, *p* < 0.001); however, overall use of corticosteroids in the placebo group increased but not significantly (30 to 50 mg/month, *p* = 0.031) (Figure 3).

A subjective assessment of AD course performed by parents in the context of AD severity, intensity of itch, and sleep disturbance also showed significant changes in the intervention group. PO-SCORAD decreased in the intervention group significantly in comparison to the placebo group, where the median difference in baseline and after 4-month intervention PO-SCORAD index was 13 for the intervention group and 1 for the placebo group (*p* < 0.001). There were also changes in the intensity of itching, sleep disturbance, and consequently on the overall quality of life (FDLQI) in the intervention group (Table 2).

Vitamin D serum values did not significantly change in both groups (intervention group: 66 to 76 nmol/L, *p* = 0.567, placebo group: 78 to 66 nmol/L, *p* = 0.211).

### Safety

During the study, there was only one adverse event: one child in the intervention group developed mild rash. As we could not exclude that this event was due to the product, we decided to exclude the patient from the trial. Other adverse effects were not reported. As usually supplements with omega-3 FA cause gastric reflux, we suggested the patients to take the received product in the evening before sleep so there was no report of gastric reflux episodes.

## 4. Discussion

We conducted a study on the efficacy of a specific blend of fish oil, DHA, EPA, GLA, and vitamin D on the severity of AD. According to the results, the severity of AD in the group receiving the active investigational product was significantly lower compared to the placebo group at the end of the 4-month treatment period (*p* < 0.001). The SCORAD index significantly decreased in children receiving the active product (Figure 1).

Moreover, the results showed a significant decrease in the use of TCS in children receiving the active product during the intervention study period—67% (from a median of 30 to 10 mg per day) (*p* < 0.001). Although the studies on omega-3 effect on atopic diseases, especially AD, are scarce, a recent study, conducted with a similar methodology to our study, has demonstrated the significant positive effect of EPA on the decrease in AD severity (SCORAD) and TCS use in children [2]. Moreover, their results were in line with ours showing the same reduction in TCS use (67%) already after 4 weeks of EPA supplementation. Also, the SCORAD index significantly decreased (49.65 ± 8.29 to 10.63 ± 9.58, *p* < 0.001).

High dosages of fish oil or omega-3 FA (EPA and/or DHA) were used in subjects with psoriasis; however, the results of these studies vary. Some studies showed significant positive effect of treatment in terms of improving itching and erythema [25,26,27], while some did not confirm it [28]. Regarding the effect of omega-3 on AD, the data are scarce; few studies performed in adults showed that dietary fish-oil supplementation may improve the biomarkers of skin inflammation and oxidative stress response [29], and that 8 weeks of DHA supplementation resulted in a significant clinical improvement of atopic eczema in terms of a decrease in SCORAD (DHA: baseline 37.0 (17.9–48.0), week 8: 28.5 (17.6–51.0), control: baseline 35.4 (17.2–63.0), week 8: 33.4 (10.7–56.2)). Moreover, this study demonstrated that the DHA group showed an increase in plasma omega-3 FA and a decrease in the omega-6 to omega-3 FA ratio [30]. However, although few, some studies did not show a significant therapeutic effect of omega-3 FA on AD which is why the question of the use of omega-3 FA as a complementary therapy is still open [18].

Our study showed a significant positive change in clinical features of AD among patients receiving the active product, consequently resulting in significant improvement in QOL. As AD symptoms (itching and sleep disturbance) improved, a subjective assessment index (family DLQI) decreased, responding to parents’ better QOL.

In children receiving placebo, there was an increase in TCS use; however, the result was not significant, and the SCORAD did not change. This was interesting, as one would expect the change in SCORAD due to TCS use. Compared to a study performed with EPA [2], where the corticosteroid use showed significant changes in SCORAD, although smaller than the changes in the group receiving EPA.

The result of our study could be related to the longer intervention period (4 months) where the dynamics of the AD were changing, even worsening depending on the season. As we enrolled most of the children during September/October/November, their AD could have flared in December/January/February due to the winter period.

From interviews with parents of AD patients, we could conclude that most of the parents did not use adequate or recommended amounts of TCS as they were expecting the positive effect of the investigational product. There was no significant change in SCORAD and PO-SCORAD index, therefore there was no significant change in other assessed parameters (itch, sleep, FLDQI).

The taste of the product was acceptable. The citrus flavor camouflaged the fish oil taste well, so most of the children accepted the product with no objections. There were no differences in adverse events between the placebo and intervention groups.

We are, however, aware of several possible limitations of the study—mainly the small number of patients enrolled, although the calculated sample size was reached, and the lack of in-clinic (on-site) follow-ups during the study period.

However, the main strength of the study is that this is, to our best knowledge, the first study of its kind. Furthermore, patients were thoroughly followed, and adherence and compliance were regularly checked by phone visits.

## 5. Conclusions

Omega-3 fatty acids in combination with GLA and vitamin D may decrease symptoms and were associated with an improved clinical picture of atopic dermatitis in children. Therefore, we can conclude that supplementation with this specific combination could be considered a safe and effective intervention that may significantly reduce the severity of AD in pediatric patients.

## Figures and Tables

**Figure 1 nutrients-16-02829-f001:**
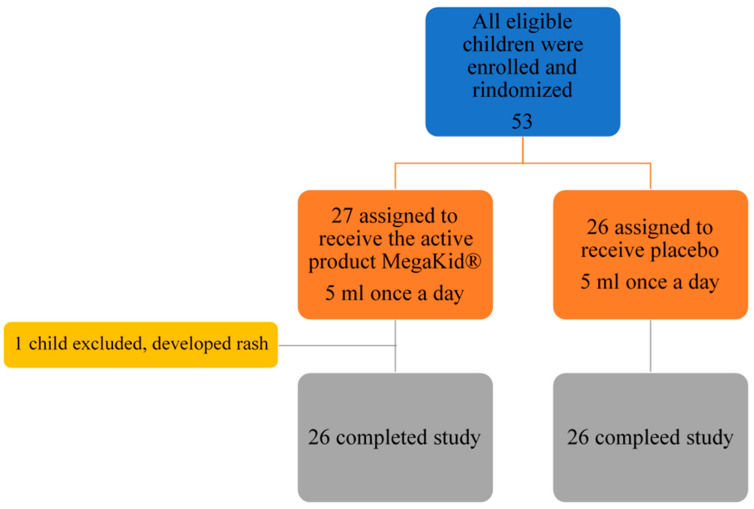
Study flow chart.

**Figure 2 nutrients-16-02829-f002:**
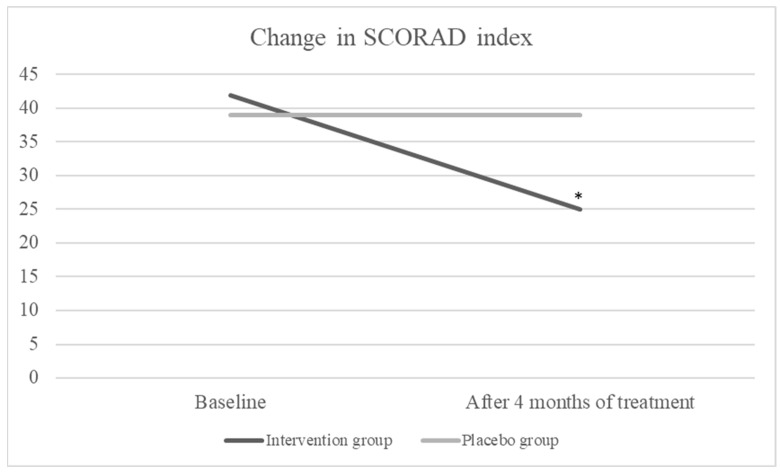
Change in SCORAD index, * *p* < 0.001.

**Figure 3 nutrients-16-02829-f003:**
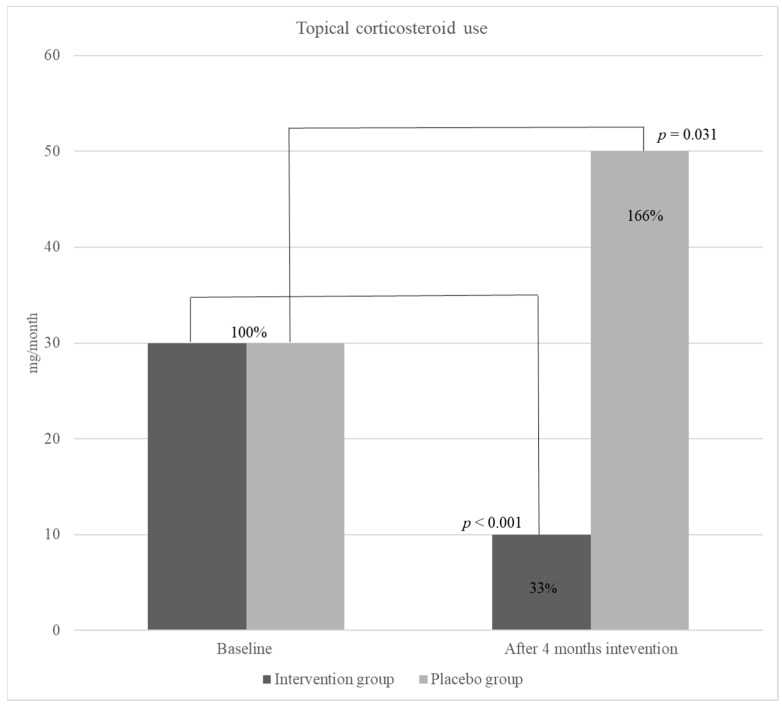
Topical corticosteroid use.

**Table 1 nutrients-16-02829-t001:** Demographic data.

	Intervention Group (n = 26)	Placebo Group (n = 26)
Age (years), median (range)	1.8 (1.1–5.9)	2.3 (1–5.7)
Sex, female N (%)	10 (38.5)	11 (42.3)
Food allergy N (%)	18 (69.2)	12 (46.2)
SCORAD N (%)		
1 mild	0 (0)	0 (0)
2 moderate	21 (80.8)	23 (88.5)
3 severe	5 (19.2)	3 (11.5)
Corticosteroid use (mg/month), median (range)	30 (0–100)	30 (0–100)
PO-SCORAD, median (range)	35 (10–51)	47 (34–65)
Itching, median (range)	8 (3–10)	8 (5–10)
Sleeping, median (range)	7 (0–10)	7 (0–10)
Supplementation vitamin D, N (%)	18 (69.2)	15 (57.7)

**Table 2 nutrients-16-02829-t002:** Difference in subjective assessment of AD severity and QOL.

Median (Range)	Intervention Group	Placebo Group	*p* Value
PO-SCORAD baseline	46	47	0.293
PO-SCORAD after 4 months of intervention	35	46	<0.001 *
Difference PO-SCORAD	13 ((−2)–29)	−1 ((−12)–20)	<0.001 *
Itching score baseline	8 (3–10)	8 (5–10)	0.653
Itching score after 4 months of intervention	4 (0–7)	8 (4–10)	<0.001 *
Itching score difference	3 (0–9)	0 ((−2)–3)	<0.001 *
Sleep disturbance baseline	7 (0–10)	7 (0–10)	0.516
Sleep disturbance after 4-month intervention	3 (0–6)	8 (0–10)	0.002 *
Sleep disturbance difference	3 (0–10)	0 ((−2)–6)	0.001 *
FLDQI baseline	17	15	0.496
FLDQI after 4 months of intervention	11	15	<0.001 *
Difference FLDQI	5 (1–14)	0 ((−3)–7)	<0.001 *

* *p* value significant < 0.005.

## Data Availability

The original contributions presented in the study are included in the article, further inquiries can be directed to the corresponding author.

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
