# Peer review of "Effect of Omega-3 Polyunsaturated Fatty Acid Supplementation on Clinical Outcome of Atopic Dermatitis in Children"

_nutrients, 2024, doi:10.3390/nu16172829_

Round 1

Reviewer 1 Report

Comments and Suggestions for Authors

The paper entitled “Effect of omega-3 polyunsaturated fatty acid supplementation on clinical outcome of atopic dermatitis in children” is informative, but there are some concerns that need to be addressed as follows.

Major Concerns

The most important hallmark of AD progression is the development of atopic itch (chronic itch and itch flares) promoted by neutrophils, basophils and CD4+T cell (Ref.).

Therefore, in addition to the clinical features of AD, the authors should confirm the positive results by examining the cellular players or serum levels of AD related cytokines (e.g. IL-4, IL-13 and IL-31). This will certainly enhance the overall quality of the manuscript.

Ref.

A basophil-neuronal axis promotes itch.

Wang F, et al. Cell. 2021 Jan 21;184(2):422-440.e17

Basophils add fuel to the flame of eczema itch.

Mali SS, Bautista DM. Cell. 2021 Jan 21;184(2):294-296.

Author Response

The paper entitled “Effect of omega-3 polyunsaturated fatty acid supplementation on clinical outcome of atopic dermatitis in children” is informative, but there are some concerns that need to be addressed as follows.

Major Concerns

The most important hallmark of AD progression is the development of atopic itch (chronic itch and itch flares) promoted by neutrophils, basophils and CD4+T cell (Ref.).

Therefore, in addition to the clinical features of AD, the authors should confirm the positive results by examining the cellular players or serum levels of AD related cytokines (e.g. IL-4, IL-13 and IL-31). This will certainly enhance the overall quality of the manuscript.

Ref.

A basophil-neuronal axis promotes itch.

Wang F, et al. Cell. 2021 Jan 21;184(2):422-440.e17

Basophils add fuel to the flame of eczema itch.

Mali SS, Bautista DM. Cell. 2021 Jan 21;184(2):294-296.

Response:

Dear Reviewer

Thank you very much for your comment.

We completely agree that it would be of great value to examine the serum levels of Ad related cytokines, however this would require additional blood sampling and cost of the study which is why in this trial we did not include the examination of inflammatory markers.

Reviewer 2 Report

Comments and Suggestions for Authors

The authors provide an interesting approach as a potential therapeutic strategy for children with AD. Although the study is clear, well-written and structured, I have some doubts and suggestions that can improve the quality of the paper. 

1. Revise all the abbreviations throughout the text. For example, describe what "GLA" stands for in the abstract.

2. Shorten the Introduction. Focus on what is known and important in this field. Use short sentences throughout the text.

3. Enhance the flow chart: use different colours and increase the font size.

4. Discuss the correlation not only with topical corticosteroids but also with antibiotics during the clinical trial. Do the authors have data on this? Did the intervention group decrease antibiotic use, or did the therapy have no influence?

5. Include this information in the Discussion section. Compare with other studies and explain how omega-3 supplementation influences or does not influence antibiotic use.

Comments on the Quality of English Language

I recommend a minor revision.

Author Response

The authors provide an interesting approach as a potential therapeutic strategy for children with AD. Although the study is clear, well-written and structured, I have some doubts and suggestions that can improve the quality of the paper.

  1. Revise all the abbreviations throughout the text. For example, describe what "GLA" stands for in the abstract.
  2. Shorten the Introduction. Focus on what is known and important in this field. Use short sentences throughout the text.
  3. Enhance the flow chart: use different colours and increase the font size.
  4. Discuss the correlation not only with topical corticosteroids but also with antibiotics during the clinical trial. Do the authors have data on this? Did the intervention group decrease antibiotic use, or did the therapy have no influence?
  5. Include this information in the Discussion section. Compare with other studies and explain how omega-3 supplementation influences or does not influence antibiotic use.

Response :

Dear Reviewer

Thank you very much for your comments.

  1. Thank you. Now we revised all abbreviations throughout text.
  2. Thank you for this comment. We now shortened the Introduction and sentences as much as we were able to do.
  3. Thank you for this comment. Please see the change in manuscript.
  4. Thank you for this comment. All our patients in both groups did not receive antibiotic therapy, therefore we did not have this as a result or comment this in Discussion section.
  5. As we did not have the patients on antibiotic therapy we do not have these results and cannot comment on effect of omega-3 FA supplementation on antibiotic use.

Reviewer 3 Report

Comments and Suggestions for Authors

The authors conducted a prospective, randomized, triple-blind, placebo-controlled parallel clinical trial aimed at evaluating the effect of omega-3 fatty acids from fish oil in combination with gamma-linolenic acid (GLA) from blackcurrant seed oil in children with atopic dermatitis (AD). Primary outcomes included changes in the severity of atopic dermatitis (AD) measured using the SCORing Atopic Dermatitis (SCORAD) and the patient-oriented SCORAD (PO-SCORAD), as well as the difference in topical corticosteroid (TCS) use. Secondary outcomes included changes in itch intensity, sleep quality, and the Family Dermatology Life Quality Index (FDLQI).

I would like to raise the following concerns.

1.

Clinical trials are generally based on the CONSORT flowchart (for placebo-controlled parallel clinical trials), but Figure 1, the study flow chart, does not seem to include this information.

2.

Randomization of the study participants (n=53) was performed using Random Allocation Software to generate randomized blocks (27 in the intervention group and 26 in the placebo group). However, 1 child was excluded due to developing a rash during the follow-up period. Data were analyzed for 52 children (26 in the intervention group and 26 in the placebo group) using a per-protocol analysis. Regardless, the authors should confirm whether the randomization of the study participants is consistent between the intention-to-treat (ITT) analysis and the per-protocol analysis.

3.

PO-SCORAD, median (range) is presented in Table 1; however, Figure 2 seems to show the mean and does not provide the standard deviation (SD). Due to this inconsistency in descriptive statistics, readers may find it difficult to interpret the results.

4.

It is suggested to present the data in a manner similar to the following formats found in N Engl J Med 2019;381:1114-1123:

FIGURE 1: Randomization and Treatment of the Participants.

TABLE 1: Characteristics of the Participants at Baseline (if randomized, P-values should not be included).

FIGURE 2: Changes in Systolic Blood Pressure and LDL Cholesterol Level at 12 Months (bars indicate the standard error).

TABLE 2: Primary and Secondary Outcomes.

This approach ensures clarity and consistency in presenting the study's methodology and results.

5.

It is also suggested to describe the statistical analysis in a manner similar to that used in N Engl J Med 2019;381:1114-1123.

6.

I do not fully understand the calculation and presentation of results in Figure 3. Topical corticosteroid use. It is suggested to provide a detailed explanation.

7.

It is suggested to provide information on the safety of the intervention.

Author Response

The authors conducted a prospective, randomized, triple-blind, placebo-controlled parallel clinical trial aimed at evaluating the effect of omega-3 fatty acids from fish oil in combination with gamma-linolenic acid (GLA) from blackcurrant seed oil in children with atopic dermatitis (AD). Primary outcomes included changes in the severity of atopic dermatitis (AD) measured using the SCORing Atopic Dermatitis (SCORAD) and the patient-oriented SCORAD (PO-SCORAD), as well as the difference in topical corticosteroid (TCS) use. Secondary outcomes included changes in itch intensity, sleep quality, and the Family Dermatology Life Quality Index (FDLQI).

 I would like to raise the following concerns.

1.Clinical trials are generally based on the CONSORT flowchart (for placebo-controlled parallel clinical trials), but Figure 1, the study flow chart, does not seem to include this information.

2.Randomization of the study participants (n=53) was performed using Random Allocation Software to generate randomized blocks (27 in the intervention group and 26 in the placebo group). However, 1 child was excluded due to developing a rash during the follow-up period. Data were analyzed for 52 children (26 in the intervention group and 26 in the placebo group) using a per-protocol analysis. Regardless, the authors should confirm whether the randomization of the study participants is consistent between the intention-to-treat (ITT) analysis and the per-protocol analysis.

3.PO-SCORAD, median (range) is presented in Table 1; however, Figure 2 seems to show the mean and does not provide the standard deviation (SD). Due to this inconsistency in descriptive statistics, readers may find it difficult to interpret the results.

  1. It is suggested to present the data in a manner similar to the following formats found in N Engl J Med 2019;381:1114-1123:

FIGURE 1: Randomization and Treatment of the Participants.

TABLE 1: Characteristics of the Participants at Baseline (if randomized, P-values should not be included).

FIGURE 2: Changes in Systolic Blood Pressure and LDL Cholesterol Level at 12 Months (bars indicate the standard error).

TABLE 2: Primary and Secondary Outcomes.

This approach ensures clarity and consistency in presenting the study's methodology and results.

  1. It is also suggested to describe the statistical analysis in a manner similar to that used in N Engl J Med 2019;381:1114-1123.
  2. I do not fully understand the calculation and presentation of results in Figure 3. Topical corticosteroid use. It is suggested to provide a detailed explanation.
  3. It is suggested to provide information on the safety of the intervention.

Dear Reviewer, thank you very much for your constructive comments.

  1. Yes, clinical trials are usually based on the CONSORT flowchart, however, as we included consecutively all eligible patients we did not count the number of non-eligible patients. So, we gave all the information on study flow that we have.
  2. Thank you for this comment. We described according to your suggestion the section on statistics: Descriptive statistics were used to describe the basic features including age, gender, duration of the intervention, and symptoms. Normality of the data distribution was analyzed with the Smirnov–Kolmogorov test. For the primary analyses, we evaluated change in severity of AD (SCORAD, PO-SCORAD) and the difference in TCS use. Due to non-normal distribution, difference was analyzed by the nonparametric Mann–Whitney test and described as median (range).

For secondary outcomes the change in itch intensity, sleep quality and FDLQI were analyzed again by the use of the nonparametric Mann–Whitney test and described as median (range).

Two-sided P values of less than 0.05 were considered to indicate statistical significance for the primary outcomes.

Statistical software SPSS (version 23; SPSS, Chicago) was used for all statistical analyses. All analyses were performed on the intention-to-treat basis, in which all of the participants in a trial are analyzed according to the intervention to which they were assigned.

  1. Figure 2. Is showing the SCORAD index median. This is also said in the text: Intention-to-treat analysis showed that after 4 months of treatment, the SCORAD index was decreased significantly in the intervention group (form median 42 to 25, p < 0.001) but there was no change in the placebo group (form median 39 to 39, p= 0.795) (Figure 2.).
  2. Thank you for your suggestions. However, all our results are shown as median not as mean so the SD bars are, of course, not included in figures. Figure 3 also shows the use of TCS as median, and not mean.

In Table 1 we deleted the last column with p values.  

As we described results and show them in tables and figures, it seems unnecessary to summarize the results once again in another table – Primary and secondary outcomes. However, to be more clear to reader we added “secondary outcomes” in the description of Table 2

  1. Changes are done according to your suggestion.
  2. Now we additionally explained the change in TCS use:

See in the method section: Use of TCS was determined by parent’s report on TCS use according the weight (mg) of the used amount of TCS per month (expressed as an absolute number).

See in the result section: The median use of TCS in the intervention group was the same as in placebo group (30 mg/month). After 4 months of intervention period, a significant decrease was detected in TCS use in the intervention group (30 to 10 mg/month, p<0.001) however, overall use of corticosteroids in the placebo group increased but not significantly (30 to 50 mg/month, p=0.031) (Figure 3.).

  1. Thank you for this suggestion. We included the paragraph on Safety in the Result section:

During the study, there was only one adverse event: one child in intervention group developed mild rash. As we could not exclude that this event was due to the product, we decided to exclude the patient form the trial. Other adverse effects were not reported. As usually supplements with omega-3 FA cause gastric reflux, we suggested the patients to take the received product in the evening before sleep so there was no report on gastric reflux episodes.

Round 2

Reviewer 1 Report

Comments and Suggestions for Authors

The authors provided informative data, but it would be preferable if publication were deferred until further studies are conducted.

Author Response

Comment 1: The authors provided informative databut it would be preferable if publication were deferred until further studies are conducted.

Reply 1: 

Dear Reviewer. To our best knowledge, there is only one study done on omega-3, concretely,  REF: doi:10.5863/1551-6776-28.1.29. where results showed positive effects on AD symptoms and lower TCS use. Therefore, we really think that this study will contribute to further research and future studies. 

Reviewer 3 Report

Comments and Suggestions for Authors

No further comment  

Author Response

Commetns1: No further comments.

Reply1: Dear Reviewer, thank you very much for your comments and contribution to improvement of the manuscript.